# Psychological Disorders and Coping among Undergraduate College Students: Advocating for Students’ Counselling Services at Kuwait University

**DOI:** 10.3390/ijerph21030245

**Published:** 2024-02-21

**Authors:** Naser M. Alotaibi, Moh A. Alkhamis, Mashael Alrasheedi, Khuloud Alotaibi, Latifa Alduaij, Fatemah Alazemi, Danah Alfaraj, Danah Alrowaili

**Affiliations:** 1Occupational Therapy Department, Faculty of Allied Health Sciences, Kuwait University, P.O. Box 31470, Sulaibekhat 90805, Kuwait; masha3l.alrashidi@gmail.com (M.A.); khuloud.mgh@gmail.com (K.A.); latifaalduaij@gmail.com (L.A.); fatma-alobody@hotmail.com (F.A.); dana.alfarj@gmail.com (D.A.); al3nziaq8@gmail.com (D.A.); 2Department of Epidemiology and Biostatistics, Faculty of Public Health, Health Sciences Center, Kuwait University, Kuwait City 13110, Kuwait; m.alkhamis@ku.edu.kw

**Keywords:** mental health, curriculum, Kuwait University, counseling center, wellness program

## Abstract

**Objectives:** The objectives of the current study are twofold. First, it aimed to explore the prevalence of depression, anxiety and stress symptoms (i.e., psychological disorders) among Kuwait University students. Second, it sought to identify and quantify the associated risk factors as well as the students’ coping strategies utilized to address these psychological disorders. **Methods:** We used a cross-sectional study at Kuwait University and selected students using a multistage stratified cluster sampling design among the 15 faculties of Kuwait University. To serve the study purposes, two cross-cultural and validated instruments were used, including the Depression, Anxiety and Stress Scale 21 (DASS-21) and the Brief-COPE scale. Descriptive statistics, as well as logistic regression analysis, were used to analyze the study findings. **Results:** A sample of 1142 students from various faculties participated in this study. We found that 681 (59.6%), 791 (69.3%) and 588 (51.5%) of the participants had depression, anxiety and stress symptoms, respectively. The highest coping strategies for stressors and challenges faced were moderate and high emotion-based coping strategies (*n* = 1063, 93.1%). Students from the Faculty of Allied Health Sciences as well as students from the Faculty of Engineering had significantly higher stress levels compared with students from other faculties (*p* < 0.05). Our results demonstrated that family problems were consistently a significant predictor of depression, anxiety and stress symptoms among Kuwait University students (*p* < 0.05). We further found that students who presented with stress and anxiety symptoms and those who practiced avoidant-focused coping strategies were substantially more likely to experience depression (ORs ≥ 2.7, *p* < 0.01). **Conclusions:** Our findings inferred that the majority of Kuwait University students have a remarkably high prevalence of mental health problems, mainly anxiety, depression, and stress symptoms along with inconsistent coping strategies toward the faced challenges during their studies. Therefore, the most important recommendation of the current study is the establishment of counselling centers in all faculties at Kuwait University. In turn, doing so facilitates the integration of wellness programs and the provision of comprehensive educational seminars, specialized training sessions and self-management techniques for Kuwait University students, leading to desired academic outcomes.

## 1. Introduction

College education and its environment are essential milestones in the lives of students that assist and guide students to work towards achieving their educational goals and career objectives [1,2]. Although the college phase is exciting for students, they often face challenges as they pursue their education, whether in their academic progress or personal or social life [3]. In addition, it is essential to note that a student’s life at the university is more emotionally and academically demanding than during any other period of their life [4]. Thus, they are exposed to a variety of hardships and struggles that make them more vulnerable to developing mental health disorders such as depression, anxiety, stress, addiction and possibly other psychiatric disorders [5,6,7]. 

The psychological effects of anxiety, stress and depression are believed to be significant indicators for the individual’s mental health; if left untreated, it could lead to adverse consequences. In particular, considering these psychological effects among university students is crucial in the contemporary world [8,9,10,11,12]. For example, anxiety and stress are common psychological impacts experienced by students that stem from academic challenges, such as exams, assignments and maintaining good grades. Because of this stress, students might experience disrupted sleep, irritability and limited concentration [13]. In addition, one of the major global concerns university students face is their high susceptibility to anxiety symptoms, which may lead to poor academic performance and diminished mental health [11]. Moreover, the pressure to achieve academic, personal and family expectations can cause some students to experience depressive symptoms, which could lead to a lack of interest in relevant daily activities as well as a lack of energy to maintain desired academic goals [9,14]. 

Furthermore, more emphasis should be placed on how these students properly adapt to these potential mental health risk factors that influence their academic performance and long-term personal goals and well-being [1,4,9,13,15]. In other words, the ability to cope with university stressors is part of students’ life and plays a vital role in ensuring that they have the ability to manage stressors that impact their performance and mental health [4]. Students differ in their ways with regard to how to address and tackle their encountered psychological disorders. For example, some students utilize positive coping strategies such as self-help, meditation and praying, effective time management, seeking help from family members and friends, and utilizing available support systems [2,16,17,18,19]. Other student, however, utilize maladaptive coping strategies that negatively influence their health and induce poor academic outcomes, including drinking, smoking, inactivity, social isolation and poor nutrition [13,16]. Understanding these associated risk factors that impact students’ psychological well-being as well as their coping strategies is critical for designing mental health intervention strategies. Past studies showed that such risk factors may include demographics (e.g., age and sex), life experiences, social connections, environmental changes, cultural context, academic setting, specialty and year of study [3,12,16,20,21,22,23]. That said, students must enhance adaptive coping strategies to meet their psychological symptoms and reach a healthy and balanced equilibrium among their academic, personal and social life aspects [10,24]. 

Understanding the epidemiology of psychological disorders among college students as well as their coping strategies when dealing with encountered challenges during their studies is invaluable, particularly on local scales. It will provide further insight into students’ life experiences and issues relevant to their academic atmosphere and learning outcomes. Therefore, the objectives of this study are targeted toward student populations affiliated with public universities and include investigating the prevalence of depression, anxiety and stress symptoms (or psychological disorders) among college students studying at Kuwait University. Additionally, we identify and quantify the associated risk factors and explore students’ coping strategies to address these psychological disorders. Hence, the present study’s findings may provide the first preliminary epidemiological picture of psychological disorders among college students in Kuwait. Therefore, such findings may help in establishing and guiding targeted mental health intervention programs aimed at the most vulnerable high-risk groups of students in a university environment.

## 2. Methods

### 2.1. Study Population and Design

We used a cross-sectional study to investigate the prevalence of stress, anxiety and depressive disorders among Kuwait University undergraduate students and their associated risk factors, as well as coping strategies. Further detailed and specific descriptions of each faculty and its specific majors can be obtained from the Kuwait University website at www.ku.edu.kw (accessed on 4 January 2024). A multistage stratified cluster sampling design was used to select our study participants from Kuwait University. Students were selected from randomly chosen faculties (i.e., clusters) from the poll of all Kuwait University Faculties. Thus, we stratified the 15 faculties of Kuwait University into two main categories, including (1) health and scientific faculties and (2) art and humanity faculties. Then, we randomly selected three faculties (or clusters) from each of the two strata. Therefore, we ended with six randomly chosen faculties, including three health and scientific faculties (the Faculty of Science, the Faculty of Allied Health Sciences and the Faculty of Engineering and Petroleum) and three art and humanity faculties (the Faculty of Education, the Faculty of Sharia and Islamic studies and the Faculty of Business Administration). The inclusion criteria were as follows: (1) all students aged 18 and older; (2) both males and females; and (3) students from first-year students to seniors were included. 

On the other hand, the exclusion criteria included all students from other faculties at Kuwait University. Our final sample size included 1142 students. We calculated the power of our sample for each faculty (i.e., cluster) and found that 44 and above participants can yield a power greater than 0.8 when the alpha is equal to 0.05 and the assumed prevalence of the outcomes is equal to or greater than 40%. 

### 2.2. Instrumentation

In our study, we utilized three components for data collection as follows:Demographic Data: it has 10 items including gender, age, marital status, socioeconomic status, nationality, medical illness, family problems, faculty, year of study and the need for counselling centers at Kuwait University.The Depression Anxiety and Stress Scale-21 (DASS-21) [25]: It includes 3 subscales of depression, anxiety and stress; each subscale includes 7 items with a total of 21 items for the whole scale. It measures the prevalence of depression, anxiety and stress symptoms. The items of the scale are rated on a four-point Likert scale, with the score for each item ranging from 0 (“does not apply to me at all”) to 3 (“applies to me most of the time”); the maximum sum for each subscale is 21. The scores for each subscale are calculated by summing the scores of the individual items then multiplying by two to calculate the final score; higher scores indicate higher psychological distress.

The scale is translated into Arabic and has sound psychometric properties [26]. We collapsed the scores of the DASS-21 into either having psychological symptoms (a score of 1) or not having psychological symptoms (a score of 0). We combined normal and mild scores as indicating no stress, anxiety or depression symptoms, whereas we combined moderate, severe, and extremely severe scores as indicating stress, anxiety or depression symptoms. Further details are illustrated in Appendix A.

3.The Brief-COPE: The Brief-COPE is a 28 item self-reported measure that evaluates coping or cognitive regulation mechanisms utilized in response to stressors [27]. The scale was cross-culturally adapted into Arabic with sound psychometric properties [28]. It consists of 14 facets (each facet contains two items) within 3 subscales:
Problem-focused coping: It relates to positive coping strategies that are adaptive and predictive of desired outcomes. It includes four facets (two items in each facet) which are as follows: (1) active coping, (2) use of informational support, (3) positive reframing and (4) planning.Emotion-focused coping: It relates to regulating emotions associated with stressful situations; it could be handled positively or negatively depending on the individual’s emotional response. It includes four facets (two items in each facet): (1) emotional support, (2) venting, (3) humor and (4) acceptance.Avoidant coping: It relates to negative coping strategies that are maladaptive and predictive of adverse outcomes. It includes six facets (two items in each facet): (1) religion, (2) self-blame, (3) self-distraction, (4) denial, (5) substance use and (6) behavioral disengagement.

Each item has responses in terms of frequency along a 4-point Likert scale as follows; 1 = I haven’t been doing this at all, 2 = A little bit, 3 = A medium amount, and 4 = I’ve been doing this a lot. The range of scoring for the subscales of problem-focused coping (8 items), emotion-focused coping (8 items) and avoidant coping (12 items) are 0–32, 0–32 and 0–48, respectively; higher scores in problem-based coping is an indication of positive (adaptive) coping strategies, whereas higher scores in avoidant coping is an indication of negative (maladaptive) coping strategies; higher scores in the emotion-based coping could be an indication of using both adaptive and/or maladaptive coping strategies based on the individual’s emotional response. Since this measure has no final score, we divided the coping scoring (responses) into three categories: low coping (0–30), medium coping (>30–70) and high coping (>70). Further details are illustrated in Appendix A. 

### 2.3. Data Collection Procedure

Ethical approval was obtained from the Institutional Review Board Health Science Center Ethical Research Committee (HSERC). Following that, students signed an in-formed consent form to ensure their understanding of the purpose of this research study as well as the confidentiality of their responses. The list of Kuwait University students from the 6 faculties chosen was obtained from the registrar’s office of faculties and/or faculty members of the assigned faculties. The online scales were self-administered and sent to all students by email, MS Teams, WhatsApp, and/or social media platforms.

### 2.4. Data Analysis

All statistical analyses were conducted in STATA© version 16. Our final outcome variables included depression, stress and anxiety statuses categorized from the total scores of the data collection instruments described above. We calculated the Cronbach alpha reliability coefficients (a) for each outcome and coping strategy to evaluate their internal consistency. All estimated a^s^ were between 0.73 and 0.88 for the subscales (Figure 1), which indicates acceptable internal consistencies [29]. We summarized our variables using frequencies and relative frequencies. We assessed their univariate relationships with each outcome using Chi-square tests (and Fisher’s exact tests when a variable had a cell count of less than 5 responses). We used logistic regression analysis for each of the three selected outcomes to model our multivariate relationships. We used a forward elimination strategy to choose our final models and assessed the statistical significance of all two-way and three-way interactions between the predictors in each multivariate model using the likelihood ratio test. Additionally, the confounding effect of non-significant variables was evaluated using the classical 10% change in the estimate method. Finally, we evaluated the goodness-of-fit of the final models using the Hosmer–Lemeshow statistic test.

## 3. Results

The outcomes and baseline characteristics of the participants are summarized in Table 1. The study findings reported that 59.6%, 69.3 and 51.5% of the participants had depression, anxiety and stress, respectively; it was found that anxiety symptoms were the most prevalent psychological symptoms among students. The estimated median scores for the selected outcomes from the subscales were 16, 17 and 20 for depression, anxiety and stress, respectively (Figure 1). At the same time, the median scores for the coping strategies were 20, 30 and 15 for problem-based, emotion-based and avoidant coping, respectively (Figure 1). Hence, the most common coping strategy utilized by students toward stressors and challenges was emotion-based coping. Furthermore, most study participants favored the establishment of multiple mental health consulting centers in all of Kuwait University faculties (n = 1036, 90.7%).

Our univariate analyses indicated that most baseline characteristics have significant relationships with the study outcomes (*p* < 0.05; Table 2), except for marital status on one side and the outcomes on the other, as well as between faculties and anxiety. Our results illustrate that over 70% of the students with depression, stress and anxiety were females aged over 20 years old (Table 2). Additionally, over 70% of the students who had any of the three study outcomes were citizens, single and from a medium socioeconomic stratum (Table 2). Also, we found that over 30% of the students susceptible to depression, stress and anxiety were from the Faculty of Allied Health Sciences. Further, our sample indicated that over 55% of the students leaned toward moderate coping strategies (Table 2). Statistics of participants without the study outcomes are summarized in Appendix A. 

Our multivariate analysis (Table 3) suggests that having family problems was consistently a significant predictor of depression, stress and anxiety (*p* < 0.05; Table 3). At the same time, for each of the two outcomes, we further found that students who were diagnosed with stress and anxiety and who practiced avoidant-focused coping strategies were substantially more likely to experience depression (ORs ≥ 2.7; Table 3). On the other hand, nationality and faculty were significant predictors of stress. Finally, medical illness and emotion-focused coping strategies were significant predictors of anxiety.

## 4. Discussion

The primary objectives of this study were to explore psychological disorders (i.e., depression, anxiety and stress) and to identify the associated risk factors as well as the coping strategies among Kuwait University students. To our knowledge, this study is the first in Kuwait, providing a preliminary epidemiological picture of local college students’ mental health status. We inferred a remarkably high prevalence of mental health problems among Kuwait University students (>50%, Table 1 and Table 2). Nonetheless, our study found that Kuwait University students experience a great deal of psychological symptoms, with anxiety being the most prevalent psychological symptom. Such findings agreed with the published literature on other parts of the world, including Asia, Africa, the Middle East, North America and South America [11]. In addition, our findings documented that depression and stress symptoms were highly prevalent among Kuwait University students, which may lead to significant implications on their academic performance and well-being; such findings are also consistent with those of the American Psychological Association [9,13,14]. That said, it is important to better understand the underlying reasons and associated factors contributing to the students’ psychological symptoms, thereby managing these symptoms appropriately [8,9,10,11,20]. Hence, this necessitates the urgent need to offer and implement invaluable mental health intervention strategies and programs to minimize these psychological symptoms accordingly [11]. Thus, doing so requires mutual efforts from university decision makers and academic staff [15].

In this study, our results indicated that female students have significantly higher psychological disorders than males, including depression, anxiety and stress symptoms (Table 2). This was also noted in the literature in different regions worldwide [16,30,31,32,33,34]. This could be attributed to genetic factors, hormonal differences, societal expectations, psychosocial issues and/or other related components [16,35,36]. As a result, we recommend hiring specialists at the university level in the area of women’s health to assess, monitor and address psychological issues relevant to females; this would eventually lead to the improvement of females’ well-being, minimizing their psychological distress and promoting desired educational outcomes among them. This study also demonstrated that academic settings and specialties contribute to psychological symptoms. In other words, our results illustrate that allied health and engineering students, particularly from the second and fourth year, are more likely to be stressed (ORs > 1; Table 3). This is because engineering students are more likely to be stressed due to the heavy academic workload, particularly in the second and fourth years of their studies, as described elsewhere [37,38]. Thus, yearly revision of students’ course load distribution, as well as other potential issues contributing to the students’ stress, are needed. 

It is worth noting that the local cultural context in Kuwait concerning women’s health, family obligations, societal expectations and personal lifestyle should be specifically considered to attain the desired mental health outcomes for women in Kuwait. To illustrate, the increased prevalence of psychological disorders as well as inconsistent coping strategies among students, particularly more with female students than male students, is attributed to cultural norms in our society [39,40]. In this regard, it is generally believed that our cultural norms prevent students from seeking mental health consulting [41]. Therefore, future studies are encouraged to further study the cultural influence on the students’ psychological disorders and coping.

Furthermore, this study demonstrated that academic settings and specialties contribute to psychological symptoms. In other words, our results illustrate that allied health and engineering students, particularly from the second and fourth year of study, are more likely to be stressed (ORs > 1; Table 3). This could be because engineering students are more likely to be stressed due to the heavy academic workload, particularly in the second and fourth years of their studies; hence, consideration of academic majors and years of study are also reported in the literature to induce stress among college students [37,38]. Therefore, to minimize stress among students in these colleges and facilitate better academic consequences, curricular changes such as review and amendment of yearly course load distribution are warranted. In contrast, students in the Faculty of Islamic Studies demonstrated that they are highly likely to exhibit depressive symptoms, which negatively influence their academic achievement. Therefore, it is significant for the students’ affairs unit in the Faculty of Islamic Studies, in collaboration with its faculty members, to scrutinize the reasons behind such symptoms, thereby providing the most appropriate mental health intervention strategies for those students. 

Moreover, our findings suggest that a higher number of allied health students were more susceptible to psychological symptoms of depression, anxiety and stress than students of other faculties; this finding is also consistent with the published literature indicating that health science students experience symptoms of anxiety and depression due to the highly competitive environment [42]. In this study, we believe this could be possibly due to the heavy course load and demand as well as working in the clinical environment with various patient populations presented with different levels of disability. Also, because allied health students have condensed courses without summer courses, this could lead to more academic stress, diminished leisure activities and limited social networking. Hence, it is well documented in the literature that increased academic stress, reduced leisure activities and limited social networking are predictive factors of adverse psychological symptoms [7,43]. Therefore, we highly encourage university decision makers to consider decreasing the students’ heavy course load and offering summer courses as strategies to manage these encountered psychological symptoms. We further recommend implementing tailored mental health consultation services, resources and support for those kinds of students across all faculties [44].

Our results demonstrated that family problems were consistently a major factor and significant predictor of depression, anxiety and stress symptoms among Kuwait University students (*p* < 0.05; Table 3). This finding was also supported in the literature. According to [45], family issues and situations contribute to burnout among nursing students, leading to negative psychological consequences. In addition, family-related problems are perceived as a critical factor causing stress among students, which could be accompanied by psychiatric disorders. Therefore, routine screening for students to determine contributing and associated factors with stress can possibly aid in early detection and management of subsequent potential complications [10,42,46]. In addition, a tense family atmosphere can negatively impact the students’ well-being, lower academic performance and likely lead to depressive symptoms [16,47,48,49].

Nevertheless, family-related problems (i.e., family conflict and communication failures within the family) further increase the prevalence of anxiety leading to poor residual academic performance [16,42,48,50]. Thus, it is highly advisable to consider the feasibility of providing comprehensive student screening, mainly to understand the students’ familial concerns, issues and support systems. Moreover, it is of value to emphasize the need to promote wellness programs on campus to address these issues that are likely influencing the students’ psychological symptoms accordingly. Doing so can improve the students’ self-efficacy toward their educational outcomes and increase the chance of academic success for them [47,50,51]. 

Furthermore, our results revealed that students with no medical illnesses were less likely to experience anxiety (OR = 0.53; 95 CI% [0.33, 0.85]; Table 3). In other words, medical illness among students was indicated as a significant predictor of anxiety. Hence, the literature similarly reported that the existence of illnesses, whether medical or mental illnesses, was a contributing factor leading to anxiety as well as psychological distress among university students [52,53,54]. Therefore, we recommend the application of medical screening of students to screen for the presence of illnesses that might hinder students’ academic performance. Thus, faculty members can encourage students to make general medical checkups to support their health and wellness during their studies. In turn, this will contribute to minimizing psychological symptoms and subsequently improve students’ academic performance. 

Concerning the coping strategies utilized by students, these were inconsistently used with a mixture of problem-focused, emotion-focused and avoidant coping strategies, with emotion-focused coping being the most used among students. Hence, our findings illustrated that emotion-focused coping and avoidant coping strategies were significant predictors of anxiety and depression, respectively (Table 3). Such practice of inconsistent or mainly negative coping strategies toward stressors could mitigate the students’ performance and further influence their well-being [55]. Therefore, students should be encouraged by the academic staff to exhibit active coping strategies, including a proactive attitude to deal with stressors, seeking help from their academic mentors whenever needed, connectedness with other students, utilizing guidance and resources available within their faculties as well as other positive coping strategies leading to improved mental health [55,56]. That said, we recommend the Deanship of Student Affairs at Kuwait University implement policies and guidelines that foster the application of psychoeducation programs in favor of supporting students’ academic motivation and desired coping strategies [57]. 

## 5. Implications and Future Directions

The current study has important implications and future directions that warrant consideration. First, as exhibited by the study findings, the high prevalence of psychological symptoms among the students is alarming, which could lead to long-term implications of mental health disorders and the butterfly effect of the consequent outcomes for those students, particularly when no intervention is provided. For example, as illustrated in our results, stress was the most significant predictor and the strongest risk factor of depression among the students (OR = 9.98; 95 CI% [6.77, 14.71], *p* < 0.01, Table 3). Therefore, we highly recommend the establishment of counselling centers in all Kuwait University faculties, mainly focusing on the integration of wellness programs, such as stress management programs; these programs are invaluable to students and play an essential role in minimizing the students’ psychological symptoms and improving their academic performance. Hence, these stress management programs can increase the students’ insight into sources and outcomes of stress, provide mindfulness skills intervention and offer cognitive behavioural therapy training, which may reduce stress levels and minimize further psychological disorders. In addition, such programs can teach the students to learn appropriate problem-solving skills and employ positive coping strategies, thereby addressing the issues of applying maladaptive coping strategies when faced with ongoing personal and academic stressors. [58,59,60,61,62]. Thus, applying stress management programs in all faculties can be a highly positive and rewarding experience. Second, in this regard, we encourage all faculties to conduct screening for all first-year students by filling out standardized measures to explore their psychological symptoms and identify the associated risk factors and coping mechanisms. Third, we strongly suggest offering a course for all Kuwait University students designed to introduce students to various stressors, whether school-related or personal, familial, psychological or environmental-related and the best strategies to positively cope with such stressors. This course should be mandatory for all students, leading to better management of encountered obstacles within the academic or non-academic environments. Doing so can also improve the student’s self-esteem, confidence level, self-efficacy, and personal achievements in current and future life events [58].

Fourth, we recommend the establishment of multiple research teams from various faculties at Kuwait University focusing on students’ mental health and wellness. The research sector at Kuwait University can offer priority grants in this area, thus motivating faculty members to contribute actively to the publication of relevant research studies. These studies can, therefore, lead to the development of updated practical and suitable programs and techniques, resulting in the desired management of psychological symptoms and effective coping strategies during their studies. Fifth, we highlight the need for conducting qualitative or mixed research studies in relevant faculties (i.e., Faculty of Islamic studies) to better understand in depth the phenomena of students’ psychological symptoms (i.e., depressive symptoms) from the students’ perspectives, thus offering the most relevant and tailored interventions accordingly. Sixth, we highly advise the development of educational initiatives throughout the educational system in Kuwait that mainly focus on addressing the psychological stressors, cognitive limitations and coping strategies among students. For example, future research studies are encouraged to conduct a similar investigation to this study by targeting psychological symptoms, associated factors and coping strategies utilized among high school students in Kuwait. This step will be an essential component for assessing and understanding the psychological symptoms and mental health issues that high school students are facing. Therefore, mental health issues can be better managed and desired educational successes can be obtained. Seventh, we encourage using cross-cultural (culturally relevant) and validated outcome measures that monitor and track the educational and therapeutic wellness programs applied to students. Lastly, a collaboration between Kuwait University and other regional and international institutions can be a significant advantage to better exchange ideas and experiences in the areas of counselling services and wellness programs, thus maximizing benefits for students’ health and well-being. 

## 6. Limitations

The current study has several limitations that deserve attention. First, the study findings have pertained to only Kuwait University students; this could yield different inferences that affect the generalizability of the study findings to students studying at other private universities in Kuwait. Therefore, we encourage replication of similar studies to be conducted among private universities’ students in Kuwait, thus grasping an informative understanding of their mental health issues and concerns. Second, while medical illness was significantly associated with psychological symptoms of Kuwait University students, the type and severity of medical illness were unknown. Therefore, future studies are recommended to inform the type and severity of the medical illnesses reported, thus enriching the study findings. Third, the type and severity of family-related problems encountered by the students were also unknown. As a result, future studies are encouraged to identify all types of familial problems and provide the degree of their severity. Consequently, a more in-depth understanding of these familial problems can be specifically addressed, and desirable learning outcomes can be achieved. Fourth, this study lacks an in-depth description of all relevant components and issues about Middle Eastern cultural norms; thus, possibly limiting its scope. Further research is therefore needed to incorporate the cultural perspective of students and the role it plays in influencing their mental health status. Despite these limitations, we strongly believe that this study’s findings, implications and recommendations are invaluable and call for better management of the challenges and psychological problems faced by students, leading to a positive educational environment in the future.

## 7. Conclusions

This study’s findings infer that the majority of Kuwait University students have a remarkably high prevalence of mental health problems, mainly anxiety, depression and stress symptoms, along with inconsistent coping strategies toward the challenges faced during their studies. In addition, further associated risk factors lead to the worsening of psychological symptoms; these risk factors mainly include stress, family problems, medical illness, type of faculty and maladaptive coping strategies. As a result, such findings negatively influence the students’ self-esteem and confidence level, leading to poor academic performance and diminished health and wellness. Therefore, the most important recommendation of the current study is the establishment of counselling centers in all faculties at Kuwait University. In turn, doing so facilitates the integration of wellness programs and the provision of comprehensive educational seminars, specialized training sessions and self-management techniques. We also recommend offering a mandatory course throughout all faculties aimed at introducing mental health education for students, understanding the associated risk factors that possibly aggravate psychological symptoms, and identifying proper coping strategies for students to meet the challenges faced during their academic journey. Moreover, we suggest establishing a local research team to study issues and strategies relevant to counselling services and wellness programs. Notably, future research pertaining to studying psychological symptoms among high school students is further encouraged to promote their health and well-being and thus prepare them positively to enter the university environment. Finally, a collaboration between Kuwait University administration/academicians with regional and international institutions in the areas of students’ counselling services, wellness programs and desired academic performance is warranted.

## Figures and Tables

**Figure 1 ijerph-21-00245-f001:**
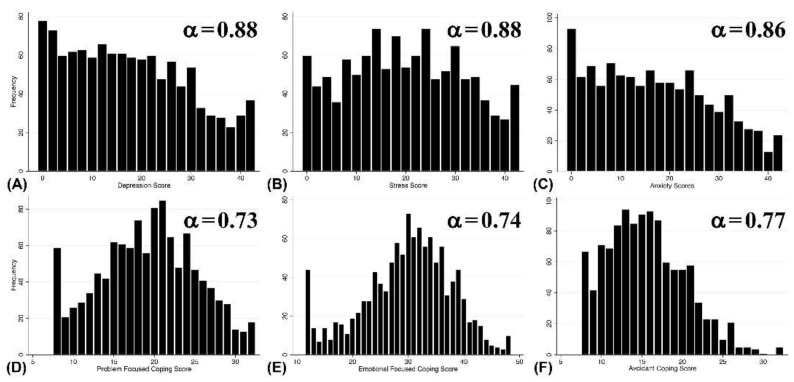
Cronbach alpha reliability coefficients (α) and median score distribution for the instrumentations’ items. (**A**) Depression, (**B**) Stress, (**C**) Anxiety, (**D**) Problem-focused coping, (**E**) Emotion-focused coping and (**F**) Avoidant coping.

**Table 1 ijerph-21-00245-t001:** Summary profile of the outcomes and baseline characteristics of the participants (n = 1142).

Variables	n (%)
**Outcomes**	
**Depression**	
Yes	681 (59.6)
No	461 (40.4)
**Stress**	
Yes	588 (51.5)
No	554 (48.5)
**Anxiety**	
Yes	791 (69.3)
No	351 (30.7)
**Baseline Characteristics**	
**Age**	
Mean ± S.D.	19.9 ± 2.1
<20 years old	498 (43.61)
>20 years old	644 (56.39)
**Sex**	
Male	303 (26.5)
Female	839 (73.5)
**Marital Status**	
Single	981 (85.9)
Married	134 (11.7)
Divorced	25 (2.2)
Widowed	2 (0.2)
**Socioeconomic Status**	
Low	148 (13.0)
Medium	891 (78.0)
High	103 (9.0)
**Nationality**	
Citizen	900 (78.8)
Resident	242 (21.2)
**Type of Faculty**	
**Health and Scientific**	623 (54.6%)
Science	140 (12.3)
Allied Health Sciences	353 (30.9)
Engineering and Petroleum	130 (11.4)
**Art and Humanity**	519 (45.4%)
Education	259 (22.7)
Sharia and Islamic Studies	116 (10.2)
Business Administration	144 (12.6)
**Year of Study**	
First	337 (29.51)
Second	323 (28.28)
Third	216 (18.91)
Fourth	266 (23.29)
**Medical Illness**	
Yes	291 (25.48)
No	851 (74.52)
**Family Problems**	
Yes	312 (27.32)
No	830 (72.68)
**Students’ opinion about the need for Consultation center at Kuwait University**	
Agree	1036 (90.7%)
Neutral	91 (8.1%)
Disagree	15 (1.4%)
**Coping Strategy**	
**Problem Focused**	
Low	106 (9.28)
Moderate	741 (64.89)
High	295 (25.83)
**Emotion Focused**	
Low	79 (6.9)
Moderate	737 (64.5)
High	326 (28.6)
**Avoidant**	
Low	180 (15.8)
Moderate	888 (77.8)
High	74 (6.5)

**Table 2 ijerph-21-00245-t002:** Univariate analysis of baseline characteristics with the study outcomes.

Characteristic	Depression	*p*-Value	Stress	*p*-Value	Anxiety	*p*-Value
	681 (59.6%)		588 (51.5%)		791 (69.2%)	
**Age**		<0.01 *		<0.01 *		<0.01 *
<20 years old	260 (38.2)		219 (37.2)		316 (39.9)	
>20 years old	421 (61.8)		369 (62.8)		475 (60.1)	
**Sex**		<0.01 *		<0.01 *		<0.01 *
Male	149 (21.9)		121 (20.6)		181 (22.9)	
Female	532 (78.1)		467 (79.4)		610 (77.1)	
**Marital Status**		0.08		0.5		0.8
Single	599 (88.0)		514 (87.4)		680 (86.0)	
Married	66 (9.7)		61 (10.4)		91 (11.5)	
Divorced	15 (2.2)		12 (2.0)		19 (2.4)	
Widowed	1 (0.2)		1 (0.2)		1 (0.1)	
**Socioeconomic Status**		0.01 *		0.01 *		0.01 *
Low	106 (15.6)		94 (16.0)		124 (15.7)	
Medium	518 (76.1)		444 (75.5)		602 (76.1)	
High	57 (8.4)		50 (8.5)		65 (8.2)	
**Nationality**		<0.01 *		<0.01 *		0.01 *
Citizen	509 (74.7)		429 (73.0)		602 (76.1)	
Resident	172 (25.3)		159 (27.0)		189 (23.9)	
**Faculty**		<0.01 *		<0.01 *		<0.01 *
Education	106 (15.6)		80 (13.6)		139 (17.6)	
Sharia and Islamic Studies	82 (12.0)		66 (11.2)		91 (11.5)	
Business Administration	80 (11.8)		73 (12.4)		103 (13.0)	
Science	85 (12.5)		69 (11.7)		92 (11.6)	
Allied Health	250 (36.7)		225 (38.3)		270 (34.1)	
Engineering and Petroleum	78 (11.5)		75 (12.8)		96 (12.1)	
**Year of Study**		0.01 *		<0.01 *		0.05
First	185 (27.2)		147 (25.0)		217 (27.4)	
Second	180 (26.4)		165 (28.1)		223 (28.2)	
Third	139 (20.4)		119 (20.2)		159 (20.1)	
Fourth	177 (26.0)		157 (26.7)		192 (24.2)	
**Medical Illness**		<0.01 *		<0.01 *		<0.01 *
Yes	222 (32.6)		205 (34.9)		251 (31.7)	
No	459 (67.4)		383 (65.1)		540 (68.3)	
**Family Problems**		<0.01 *		<0.01 *		<0.01 *
Yes	267 (39.2)		238 (40.5)		281 (35.5)	
No	41 (60.8)		350 (59.5)		510 (64.5)	
**Coping Strategy**						
Problem Focused		<0.01 *		<0.01 *		<0.01 *
Low	25 (3.7)		19 (3.2)		33 (4.2)	
Moderate	446 (65.5)		360 (61.2)		510 (64.5)	
High	210 (30.8)		209 (35.5)		248 (31.3)	
Emotion Focused		<0.01 *		<0.01 *		<0.01 *
Low	11 (1.6)		8 (1.4)		17 (2.2)	
Moderate	423 (62.1)		346 (58.8)		497 (62.8)	
High	247 (36.3)		234 (39.8)		277 (35.0)	
Avoidant		<0.01 *		<0.01 *		0.04 *
Low	33 (4.9)		24 (4.1)		47 (5.9)	
Moderate	577 (84.7)		498 (84.7)		672 (85.0)	
High	71 (10.4)		66 (11.2)		72 (9.1)	

* Statistically significant.

**Table 3 ijerph-21-00245-t003:** Multivariate logistic regression models for study outcomes and their associated risk factors.

Variable	Depression Model ^a^			Stress Model ^b^			Anxiety Model ^c^		
	ORs ^d^	95% CI ^e^	*p*-Value	ORs	95% CI	*p*-Value	ORs	95% CI	*p*-Value
Sex									
Female	1.25	(0.81, 1.90)	0.3	1.29	(0.82, 2.02)	0.26	1.21	(0.83, 1.76)	0.317
Marital Status				-	-	-	-	-	-
Married	0.51	(0.28, 0.92)	0.02 *						
Divorced	1.22	(0.40, 3.68)	0.72						
Widowed	0.53	(0.00, 55.63)	0.85						
Nationality									
Resident	-	-	-	1.65	(1.10, 2.57)	0.03 *	-	-	-
College							-	-	-
Islamic Studies	2.17	(1.20, 4.33)	0.03 *	1.3	(0.64, 2.62)	0.45			
Business	0.68	(0.37, 1.26)	0.23	1.56	(0.81, 3.00)	0.18			
Science	1.54	(0.84, 2.83)	0.15	1.29	(0.68, 2.44)	0.43			
Allied Health	1.5	(0.92, 2.45)	0.09	2.39	(1.43, 3.98)	<0.01 *			
Engineering	0.86	(0.45, 1.63)	0.66	2.89	(1.49, 5.61)	<0.01 *			
Year of Study							-	-	-
Second	0.76	(0.48, 1.19)	0.24	1.59	(1.08, 2.53)	0.04 *			
Third	1.22	(0.73, 2.04)	0.43	1.14	(0.67, 1.93)	0.63			
Fourth	1.6	(0.98, 2.61)	0.06	1.64	(1.03, 2.68)	0.05 *			
Family Problems									
No	0.35	(0.22, 0.55)	<0.01 *	0.58	(0.38, 0.89)	0.01 *	0.55	(0.33, 0.92)	0.02 *
Medical illnesses									
No	-	-	-	-	-	-	0.53	(0.33, 0.85)	0.01 *
Stress									
Yes	9.98	(6.77, 14.71)	<0.01 *	5.05	(3.48, 7.34)	<0.01 *	6.92	(4.38, 10.93)	<0.01 *
Anxiety									
Yes	3.25	(2.15, 4.89)	<0.01 *	3.25	(7.16, 15.81)	<0.01 *	5.3	(3.59, 7.82)	<0.01 *
Coping Strategy									
Emotional Focused									
Moderate	1.74	(0.69, 4.40)	0.23	-	-	-	3.17	(1.58, 6.35)	0.03 *
High	1.85	(0.68, 5.02)	0.22				4.83	(2.24, 10.40)	0.01 *
Avoidant									
Moderate	2.73	(1.57, 4.75)	<0.01 *	1.84	(0.95, 3.56)	0.06	-	-	-
High	11.22	(2.63, 47.84)	<0.01 *	2.55	(0.83, 7.81)	0.09			

Hosmer–Lemeshow goodness-of-fit *p* = ^a^ 0.72, ^b^ 0.81, ^c^ 0.62; ^d^ odds ratio; ^e^ 95% confidence interval; * statistically significant.

## Data Availability

Data can be obtained through contact with the corresponding author.

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
