# Peer review of "Psychological Disorders and Coping among Undergraduate College Students: Advocating for Students’ Counselling Services at Kuwait University"

_ijerph, 2024, doi:10.3390/ijerph21030245_

Round 1

Reviewer 1 Report

Comments and Suggestions for Authors

Thank you for this paper, it shows interesting results that uncover the specifics of educational situation in Middle East university, which is very important since most psychological results originate from Western studies and it is unclear if they are fundamental or specific to Western culture.

Still, there are some comments to be addressed.

1. In line 106 you state that Kuwait university has 15 faculties/ colleges, but than in lines 108 and 110 their amount increase to 16. How many faculties are there?

2. In your analysis you have variable "family problems", but in methods section I couldn't find any description of this variable. You have students, most of them are single, so the family problems were related to their parent families? On the other hand, some of them are married, and having family at that young age together with attempts to get a degree might also cause some tension. So it would be reasonable to explain what kind of problems were addressed in your study.

3. Even though it is obviously necessary to obtain data about psychological characteristics of Kuwait students, still all the results are consistent with what is already known and was highly predictable. And there are no psychological or cultural variables that would underline specifics of these processes among Kuwait students. So my sugestion would be  to give more cultural specifics in the paper, cultural background, like to what extent it is common to go to a university - is it a usual choice for a school graduate or it is highly competitive and needs lots of effors, it is also interesting that you have significantly more girls than boys - is it relevants to university demographics? If so - maybe some background would help as in most western countries humanitarian and social sciences have more girls, while technical sciences have more boys and when it gets to average - they are 50/50.

Comments on the Quality of English Language

Ehglish needs minor editing throughout the text, for example for articles and grammar.

Reviewer 2 Report

Comments and Suggestions for Authors

Thank you for inviting me to review this paper. This is a well-written paper. However, the following issues should be addressed to further improve the manuscript.

1. In section 2.1, I do not think the description of Kuwaiti University is necessary.

2. Sample bias analysis should be conducted to ensure the sample can represent the university’s student population.

3. Line 144-190, it is better for authors to display the three subscales  in a table for an easy read.

4. Table 2 missed three columns: the students without depression, stress, and anxiety.

5. Table 3 could be organized better. Depression, Stress, and Anxiety could be in three columns. Also, in Table 3, “risk factor” is not appropriate, instead, “variables” are recommended.

6. In table 3, the rows of reference groups can be removed. Since the authors have Table 1, readers should know the omitted one is the reference group.

7. Since logistic models include individual characteristics, thus, the authors also should illustrate somewhere in the paper why they include sex, marital status, etc. in their model. Have previous studies found that sex, marital status, and college major fields will influence their depression, stress, and anxiety?

8. In section 5, the present study tries to identify what factors that will influence students’ depression, stress, and anxiety. However, in this section, the authors try to illustrate how counseling program can alleviate depression, stress, and anxiety. In other words, the recommendations are not aligned with the findings.  

9. The title should be shortened.

Minor issues:

1. p value: p should be lower case and italic, p, also change “p-values” to “p”, for example p < 0.05   

2. The abstract is not required to be structured.

3. The reference style should follow MDPI journals’ requirements.  

Round 2

Reviewer 2 Report

Comments and Suggestions for Authors

I appreciate the authors' substantial revisions. The manuscript is much improved. However, as mentioned in the authors' response letter, I believe a new question MUST be addressed; that is the authors should clearly state why they included different sets of variables in the three models in Table 3. To be more transparent, I think that organizing models of Depression, Stress, and Anxiety in three columns would make it clearer to readers which variables were not included in which models. 
